# Structural basis of *Blastomyces* Endoglucanase-2 adjuvancy in anti-fungal and -viral immunity

**Lucas dos Santos Dias**[1], **Hannah E. Dobson**[1], **Brock Kingstad Bakke**[2], **Gregory C. Kujoth**[1], **Junfeng Huang**[3], **Elaine M. Kohn**[1], **Cleison Ledesma Taira**[1], **Huafeng Wang**[1], **Nitin T. Supekar**[4], **J. Scott Fites**[1], **Daisy Gates**[2], **Christina L. Gomez**[5], **Charles A. Specht**[5], **Stuart M. Levitz**[5], **Parastoo Azadi**[4], **Lingjun Li**[3], **Marulasiddappa Suresh**[2], **Bruce S. Klein**[1,6,7], **Marcel Wüthrich**[1] *

1 Department of Pediatrics, University of Wisconsin School of Medicine and Public Health, University of Wisconsin-Madison, Madison, Wisconsin, United States of America, 2 Department of Pathobiological Sciences, School of Veterinary Medicine, University of Wisconsin-Madison, Madison, Wisconsin, United States of America, 3 School of Pharmacy, University of Wisconsin-Madison, Madison, Wisconsin, United States of America, 4 Complex Carbohydrate Research Center, University of Georgia, Athens, Georgia, United States of America, 5 Department of Medicine, University of Massachusetts Medical School, Worcester, Massachusetts, United States of America, 6 Department of Internal Medicine, University of Wisconsin School of Medicine and Public Health, University of Wisconsin-Madison, Madison, Wisconsin, United States of America, 7 Deparament of Medical Microbiology and Immunology, University of Wisconsin School of Medicine and Public Health, University of Wisconsin-Madison, Madison, Wisconsin, United States of America

* mwuethri@wisc.edu

**Data Availability Statement:** All relevant data are within the manuscript and its supporting information files.

## Abstract

The development of safe subunit vaccines requires adjuvants that augment immunogenicity of non-replicating protein-based antigens. Current vaccines against infectious diseases preferentially induce protective antibodies driven by adjuvants such as alum. However, the contribution of antibody to host defense is limited for certain classes of infectious diseases such as fungi, whereas animal studies and clinical observations implicate cellular immunity as an essential component of the resolution of fungal pathogens. Here, we decipher the structural bases of a newly identified glycoprotein ligand of Dectin-2 with potent adjuvancy, *Blastomyces* endoglucanase-2 (Bl-Eng2). We also pinpoint the developmental steps of antigen-specific CD4⁺ and CD8⁺ T responses augmented by Bl-Eng2 including expansion, differentiation and tissue residency. Dectin-2 ligation led to successful systemic and mucosal vaccination against invasive fungal infection and Influenza A infection, respectively. O-linked glycans on Bl-Eng2 applied at the skin and respiratory mucosa greatly augment vaccine subunit- induced protective immunity against lethal influenza and fungal pulmonary challenge.

## Author summary

Fungal disease remains a challenging clinical and public health problem in part because there is no commercial vaccine available. The lack of suitable adjuvants is a critical barrier

**Funding:** The study was supported by the National Institutes of Health grants AI093553 (MW), AI035681 (BK), AI040996 (BK), AI124299 (MS), AI149793 (MS), AI025780 (SL), U01CA231081 (LL) and RF1AG052324 (LL). The Orbitrap instruments were purchased through the support of an NIH shared instrument grant (NIH-NCRR S10RR029531 to LL) and the University of Wisconsin-Madison, Office of the Vice Chancellor for Research and Graduate Education with funding from the Wisconsin Alumni Research Foundation. The glycan analysis was supported in part by National Institutes of Health grant (S10OD018530) (PA). The funders had no role in study design, data collection and analysis, decision to publish, or preparation of the manuscript.

**Competing interests:** The authors have declared that no competing interests exist.

to developing safe and effective vaccines against fungal pathogens. Current adjuvants such as alum preferentially induce antibody responses which may be limited in mediating protection against fungi. Clinical observations and animal studies implicate cellular immunity as the essential component for the resolution of fungal infections. We have recently discovered an adjuvant that augments cell mediated immune responses and vaccine induced protection against fungi. Here, we identified the structural and mechanistic requirements by which this newly discovered adjuvant induces cell mediated immunity against fungi. As a proof of principle we also demonstrate that the adjuvant drives cellular immune responses against viruses such as influenza. We anticipate that our adjuvant can be used for vaccination with safe subunit vaccines against many microbial pathogens including viruses, intracellular bacteria, fungi and parasites that require cell mediated immune responses.

## Introduction

The lack of appropriate adjuvants is one major impediment to developing safe and effective vaccines against infections with viral and fungal pathogens. Currently, there are no effective vaccines against fungi and respiratory viruses available, including broadly protective vaccines against seasonal influenza A viruses (IAV). IAV vaccines licensed in the USA rely on the generation of neutralizing antibodies targeting hemagglutinin (HA) antigens, which is the most frequently mutated IAV protein [1]. Thus, vaccine antigens must be adjusted annually to match the HA predicted for the next influenza season. In addition, humoral immune responses tend to be short lived, and provide inconsistent cross-protection against heterosubtypic and heterotypic viruses [1–4]. Unlike most neutralizing antibodies, cellular immunity mediated by CD8$^+$ cytotoxic T lymphocytes (CTLs) target structural IAV epitopes such as nucleoprotein peptides. These targets are less mutable and broadly conserved, and they generate long-lived memory cells capable of mounting cross-protective recall responses to heterosubtypic influenza infection [5–8].

Clonally-derived, adoptively transferred monoclonal antibodies may confer protection against some fungi [9–12], but the contribution of antibody to host defense is limited. Animal studies and clinical observations have implicated cellular immunity as an essential component for the resolution of most fungal infections [13–18]. Vaccine-induced resistance to fungi requires CD4$^+$ T cells that produce the pro-inflammatory cytokines IL-17 (Th17 cells) and IFN-γ (Th1 cells) [13, 15, 19].

We recently demonstrated that fungal recognition by C-type lectin receptor (CLR) Dectin-2 is required for the differentiation of protective, antifungal Th1 and Th17 cells [20]. Dectin-2 is expressed by macrophages, dendritic cells (DCs), neutrophils, monocytes, and other myeloid cells that binds high-mannose structures displayed on the surface of microorganisms, particularly fungi [21, 22]. The extracellular carbohydrate-recognition domain (CRD) of Dectin-2 shows high affinity for binding α-1,2 and α-1,4 mannose in a Ca$^{2+}$-dependent fashion [23] and, together with FcRγ, triggers intracellular signaling [24]. Few Dectin-2 ligands have been described, including MP98, a protein of *Cryptococcus* that is highly modified by N- and O-linked glycans [25–27]; a mannosylated O-antigen of *Hafnia alvei* [28]; an O-linked mannobiose-rich glycoprotein from *Malassezia* [29]; a mannose-capped lipoarabinomannan (Man-LAM) of *Mycobacterium tuberculosis* [30]; and *Blastomyces* endoglucanase-2 (Bl-Eng2), a cell wall glycoprotein from *B. dermatitidis* that was recently discovered in our lab [27].

Dectin-2 ligands share the presence of mannose in their structure, but sharp differences are evident. Man-LAM and the ligand from *H. alvei* have α-1,2 mannose residues attached to a lipid domain, whereas MP98 and Bl-Eng2 exhibit a protein backbone with N- and O-glycans attached to specific anchor amino acids. The precise structure of the glycoprotein ligand for Dectin-2 is debated. While many CLRs recognize glycans or glycolipids, some recognize protein ligands. For instance, Mincle recognizes SAP30, a protein released by dead cells [31]. However, Dectin-2 has also been reported to recognize N-linked high-mannose structures such as Man9GlcNac2 [21] and O-linked glycans such as α1,2 mannose [29, 30, 32].

Man-LAM from mycobacteria activates BMDCs to produce pro- and anti-inflammatory cytokines and promotes antigen-specific T cell responses in a Dectin-2 dependent manner [30]. Man-LAM as an adjuvant in mice induced experimental autoimmune encephalitis (EAE) mediated by Th17 cells. Similarly, the use of fungal glycosylation to provide N- and O-linked mannosylation from cryptococcal MP98 [26, 33] increased the capacity of the model antigen OVA to stimulate Ag-specific T cell responses [34]. We have reported that the Dectin-2 ligand Bl-Eng2 exerts potent adjuvant properties, inducing Ag-specific Th17 and Th1 cells, augmenting activation and killing of fungi by myeloid cells, and protecting mice from lethal fungal infection [27].

Here, we sought to address three questions: (i) the structural basis for Bl-Eng2 adjuvant activity–specifically the contribution of the protein backbone vs. the N-linked or O-linked glycans; (ii) its action on CD4$^+$ and CD8$^+$ T cell subsets; and (iii) its ability to augment cellular immunity to fungal and viral pathogens of the respiratory tract. We found that O-linked α1,2 mannosylation of Bl-Eng2 is chiefly recognized by Dectin-2 and augments the expansion, differentiation, tissue residency of Ag-specific CD4$^+$ and CD8$^+$ T cells in the lung and protects mice against fungal and viral infection.

## Results

### Dectin-2 recognizes mannose residues in Bl-Eng2, but not the protein backbone

Many CLRs recognize glycans or glycolipids, but some recognize protein ligands. For instance, Mincle recognizes SAP30, a protein released by dead cells [35]. Moreover, we observed that digestion of the crude, water-soluble, cell wall extract (CWE) from *Blastomyces* vaccine yeast with either proteinase K or endo-mannosidases reduced Dectin-2 recognition as measured by reduced ligand activity by corresponding reporter cells [27], suggesting that both the protein and glycan moieties may contribute.

To test whether the protein moiety of Bl-Eng2 harbors ligand activity, we took two approaches: first, we tested if the recombinant, glycan-free Bl-Eng2 protein backbone is sufficient for the recognition by Dectin-2. Bl-Eng2 expressed in *E. coli* (lacking glycosylation) was not recognized by Dectin-2 reporter cells, whereas Bl-Eng2 expressed by *Pichia* triggered reporter activity (**Fig 1A and 1B**). In a second approach, we chemically deglycosylated *Pichia* expressed Bl-Eng2 (displays both *N*- and *O*-linked glycans) with trifluormethanesulfonic acid (TFMS) (**Fig 1C and 1D**). TFMS treatment reduced the molecular weight of Bl-Eng2 (**Fig 1C**) and abolished recognition by Dectin-2 reporter cells (**Fig 1D**). Conversely, *Pichia* expressed Bl-Eng2 digested with proteinase K retained Dectin-2 reporter activity. Thus, Bl-Eng2 glycans, but not the protein backbone itself, are responsible for the ligand activity. Moreover, the results with proteinase K treatment suggest that the protein backbone and tiertiary structure are dispensable for Dectin-2 recognition of the glycans.

Dectin-2 is known to recognize mannose residues, specifically the disaccharides Manα1-2Man and Manα1–4 that are present on the surface of microbes. Dectin-2 recognizes these

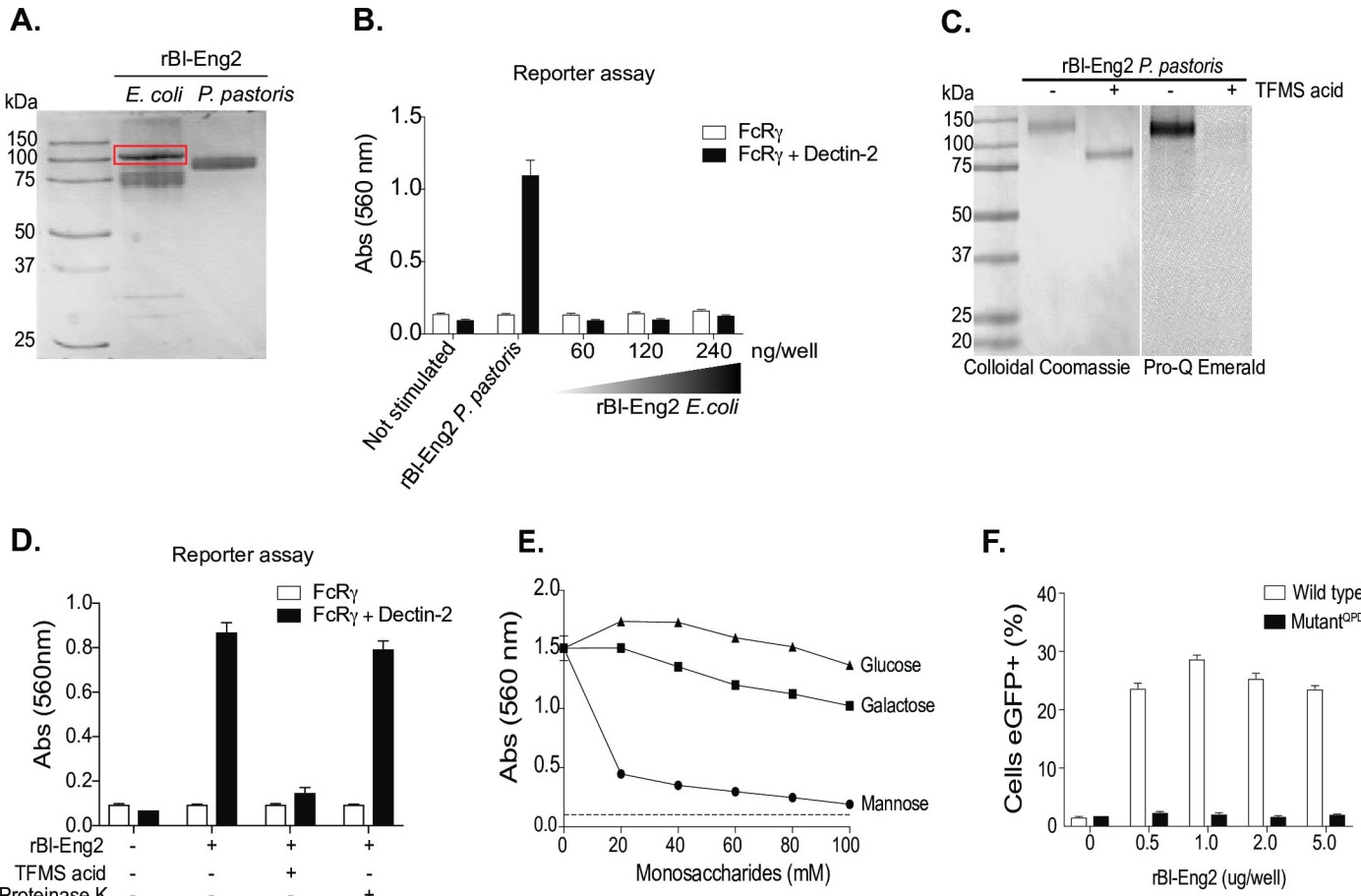

**Fig 1. Dectin-2 recognizes mannosylation but not the protein backbone of Bl-Eng2. A)** Recombinant (r) Bl-Eng2 expressed in *E. coli* (the higher molecular band in the red rectangular box was confirmed to be Bl-Eng2 by western blot) or *P. pastoris* and analyzed by SDS-PAGE. **B)** B3Z reporter cells expressing FcγR plus Dectin-2 or FcγR (B3Z) alone (negative control) were incubated with rBl-Eng2 from *P. pastoris* (30 ng) or *E. coli* (30-240ng) and reporter activity measured. **C)** rBl-Eng2 from *P. pastoris* was deglycosylated with trifluormethanesulfonic acid (TFMS) and analyzed by SDS-PAGE. Colloidal Coomassie stain was used to evaluate the molecular size and Pro-Q Emerald 300 to evaluate glycosylation. **D)** Deglycosylated and proteinase K-treated rBl-Eng2 from *P. pastoris* was analyzed by Dectin-2 reporter assay. **E)** rBl-Eng2 from *P. pastoris* was tested for Dectin-2 reporter activity in the presence of soluble glucose, galactose or mannose. **F)** NFAT-GFP reporter cells expressing wild-type Dectin-2 or a mutant CRD that doesn't recognize mannose were stimulated with rBl-Eng2 and analyzed for GFP expression by flow cytometry. The data are representative of three independent experiments.

and other ligands by the extracellular C-terminal C-type carbohydrate-recognition domain (CRD) of dectin-2 that contains the three amino acid sequence, EPN. We investigated whether free mannose blocks Dectin-2 reporter recognition of Bl-Eng2. Soluble mannose reduced reporter recognition of Bl-Eng2 in a concentration dependent manner (**Fig 1E**), suggesting that mannose is critical for Dectin-2 recognition. In a second approach, we used Bl-Eng 2 to stimulate reporter cells carrying a CRD mutation (QPD) in Dectin-2. Mutant reporter cells failed to respond, suggesting that mannose residues in Bl-Eng2 are recognized by this CLR (**Fig 1F**).

## Bl-Eng2 harbors N-linked and O-linked glycosylation sites

To define the identity of glycans associated with the protein backbone of Bl-Eng2, we digested the glycoprotein with multiple enzymes and performed liquid chromatography-mass spectrometry (LC-MS) based glycopeptide identification and characterization. We then combined

the LC-MS results from chymotrypsin, trypsin and Glu-C digestion for a comprehensive profiling of the glycosylation states of Bl-Eng2. We identified four glycan types: N- and O-linked glycans with and without phosphorylation (**Fig 2A**). The O-linked glycans are a series of 1 to 30 mannose monomers and the N-linked glycans are composed of two N-acetyl-glucosamine (GlcNAc) moieties coupled to a series of 9 to 20 mannose residues. The Bl-Eng2 protein is glycosylated at multiple locations along the entire amino acid sequence (**Fig 2B**). The O-linked mannose residues at the N-terminus, including the catalytic GH16 domain and surrounding sequence, are short and range from 1 to 3 mannoses. In contrast, the O-linked glycans at the C-terminus are generally longer and have sizes ranging between 2 and 21 mannose residues. Some O-glycans at the C-terminus were also phosphorylated. Full MALDI TOF/TOF mass spectrum analysis of permethylated N and O-linked glycans of Bl-Eng2 released by hydrazinolysis confirmed the glycopeptide analysis. The Ser/Thr rich area near the C-terminus contains 68 predicted O-linked glycosylation sites (**S1 Fig**). We were unable to digest the Ser/Thr rich region with the various enzymes we used, and thus could not experimentally verify the predicted mannosylation in this region. In addition to the O-glycans, we found 4 N-linked glycosylation sites at the N-terminus; the glycan at site N109 was phosphorylated (**Fig 2B**).

Each site-specific glycosylation pattern annotation is supported by high-resolution tandem mass spectra acquired using a hybrid fragmentation technique called Electron Transfer/ Higher-Energy Collision Dissociation (EThcD). EThcD enables peptide backbone fragmentation while preserving the more labile glycosidic bond, thus allowing the site of glycosylation and identity of glycan composition to be determined. In **Fig 2C**, we provide an example of the EThcD tandem mass spectrum annotation for the O-linked glycopeptide (QKLIS$_{Hex(7)phospho}$EE), Hex = Mannose. The tandem mass spectrum is annotated with a series of sequence-specific b and y ions indicative of amino acid sequence, whereas c and z ions generated via ETD fragmentation suggest the location of glycosylation site at Ser. HCD as a supplemental fragmentation also produces a series of glycan cleavage products annotated as sequential losses of mannose residues (-162 Da), represented by green circles. Two adjacent peaks with identical number of mannose residues with 80 Da mass difference suggest the loss of phosphorylation, which modifies the 2$^{nd}$ mannose residue. Furthermore, Y-ions (derived from the intact peptide + glycan remnant peaks) generated by HCD fragmentation allows annotation of glycan composition and structure (illustrated along the top of the panel). The B-ions are the oxonium ions generated from glycan cleavage by HCD, suggesting the presence of various glycan moieties; these are annotated in green, in the low *m/z* region.

To investigate the linkage of the O-linked mannose residues, we performed GC-MS analysis. We found that most of the mannose was 1,2 linked (64.6%) and the rest were mostly terminal mannose linkages (32.4%) (**Fig 2D**). We also found trace amounts of 1,4; 1,6; 2,3 and 2,6 linked mannose. In summary, Bl-Eng2 is heavily decorated with longer chains of 1,2 O-linked mannose at the C-terminus and short, terminal O-linked mannose and a few N-linked mannose residues at the N-terminus.

## Dectin-2 senses Bl-Eng2's C-terminal O-linked glycans, but not the N-terminal N-linked glycans

To investigate the type of mannans on Bl-Eng2 that are sensed by Dectin-2, we took two approaches: First, we treated full length rBl-Eng2 and the C-terminal-truncated version (ΔC rBl-Eng2) that lacks the Ser/Thr rich region (marked red in Figs 2B and 3A) with PNGase F. This enzyme specifically cleaves off N-linked glycans from the protein core. PNGase F treatment did not alter the molecular weight of full length Bl-Eng2 nor ΔC rBl-Eng2 (indicated by red arrows) (**S2A and S2B Fig**), nor did it affect recognition by Dectin-2 (**S2C Fig**). The

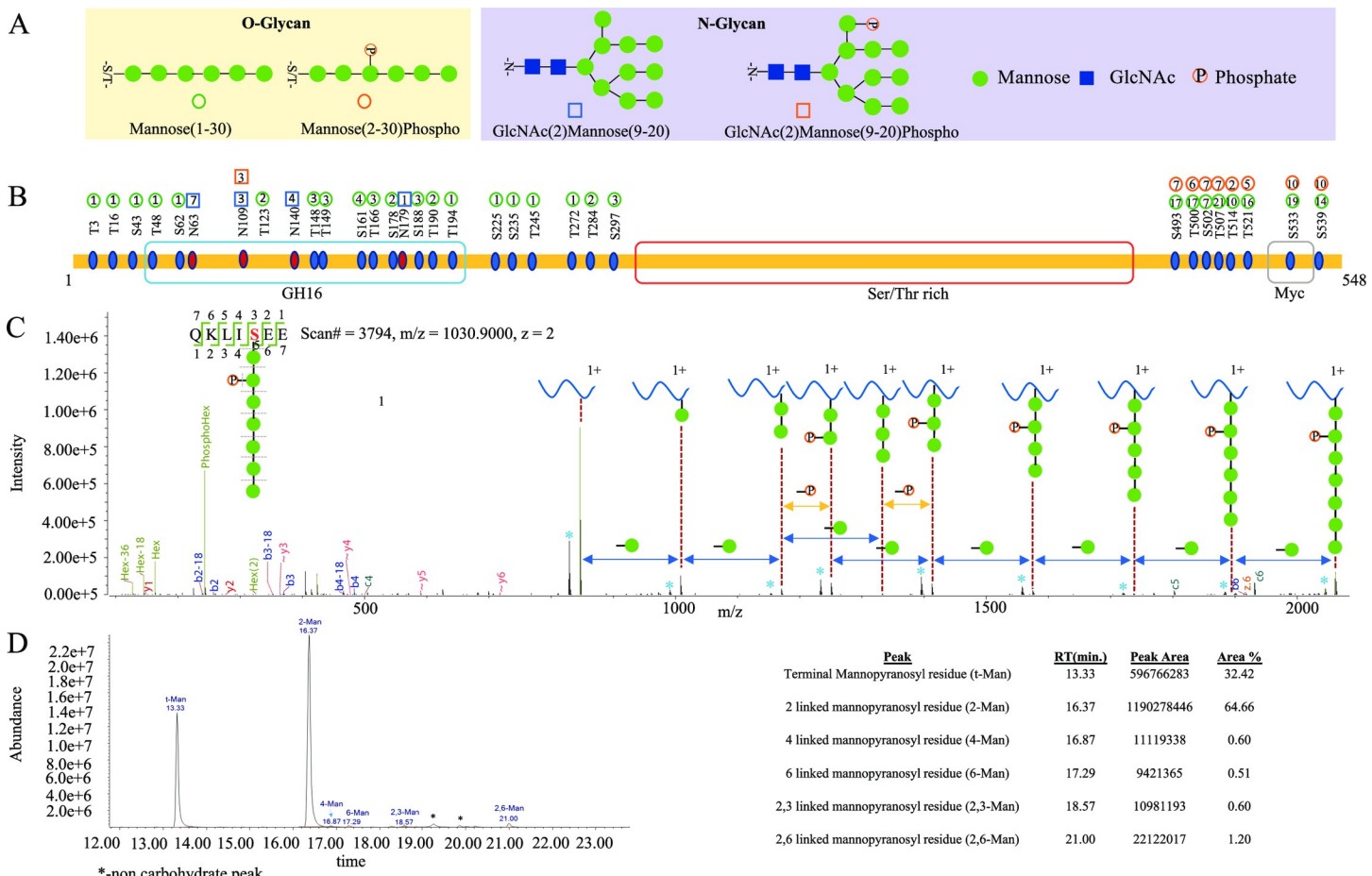

**Fig 2. Glycosylation profiling of Bl-Eng2. (A)** Glycan modificatons of Bl-Eng2 identified included N-linked (indicated by square) and O-linked glycans (indicated by circle) with (shown as red with P) and without phosphorylation. **(B)** Glycosylation sites and glycoform annotation of rBl-Eng2. The O-linked mannose residues at the N-terminus, including the catalytic GH16 domain and surrounding sequence, are short and range from 1 to 3 mannoses. The number in the circle indicates the number of mannose residues. Green circle means mannose, red circle means phosphorylated mannose, with the number in red circle indicating the position of mannose being phosphorylated. As shown, the O-linked glycans at the C-terminus are generally longer and have size ranging between 2 and 21 mannose residues. Glycosylation of the Serine/Threonine rich region was not profiled due to lack of enzyme cleavage sites. **(C)** Annotated electron transfer/higher-energy collision dissociation (EThcD) tandem mass spectrum of a representative O-linked glycopeptide (QKLIS$_{Hex(7)phospho}$EE) of rBl-Eng2, Hex = Mannose. Singly charged Y-ions (peptide plus glycan remnant) are annotated along the top. Blue asterisks (*) represent the deamination peaks. A series of sequence-specific b (labeled in blue) and y (labeled in red) type of fragment ions enable derivation of amino acid sequence of the peptide. A series of sequential loss of mannose peaks (-162 Da, indicated by green circles between two dashed lines above the blue double arrowhead) suggests a total of 7 mannose residues modified at the serine residue. Two adjacent peaks with identical number of mannose residues with 80 Da mass difference (shown as–P above the yellow double arrowhead) suggest the loss of phosphorylation, which modifies the 2nd mannose residue attached to the serine residue of the O-glycopeptide. The oxonium ions generated from glycan cleavage by HCD suggest the presence of various glycan moieties, annotated in green, in the low *m/z* region. **(D)** Glycan linkage information revealed by GCMS analysis. 1,2-linked mannose and terminal mannose are the major linkages of the glycans, with trace amounts of 1,4-linked Mannose, 1,6-linked mannose, 2,3-linked mannose and 2,6-linked mannose. Relative percentages of these glycans are shown on the right side of the panel.

control glycoprotein RNaseB was digested by PNGase F (indicated by green arrows) showing that the enzyme was active. Full length rBl-Eng2 and ΔC rBl-Eng2 devoid of N-linked glycans stimulated Dectin-2 reporter cells to the same extent as untreated Bl-Eng-2 proteins. Thus, N-glycosylation of Bl-Eng2 is dispensable for Dectin-2 recognition.

In a second approach, we assayed truncated rBl-Eng2 proteins that lack either the O-linked glycan rich C-terminus (ΔC rBl-Eng2) or the N- and O-linked N terminus (ΔN rBl-Eng2) (**Figs 3A and S3**). Removal of the C terminus reduced the molecular weight of rBl-Eng2 from 120kDa to 37kDa (**Fig 3B),** supporting the notion that the C terminus is heavily glycosylated

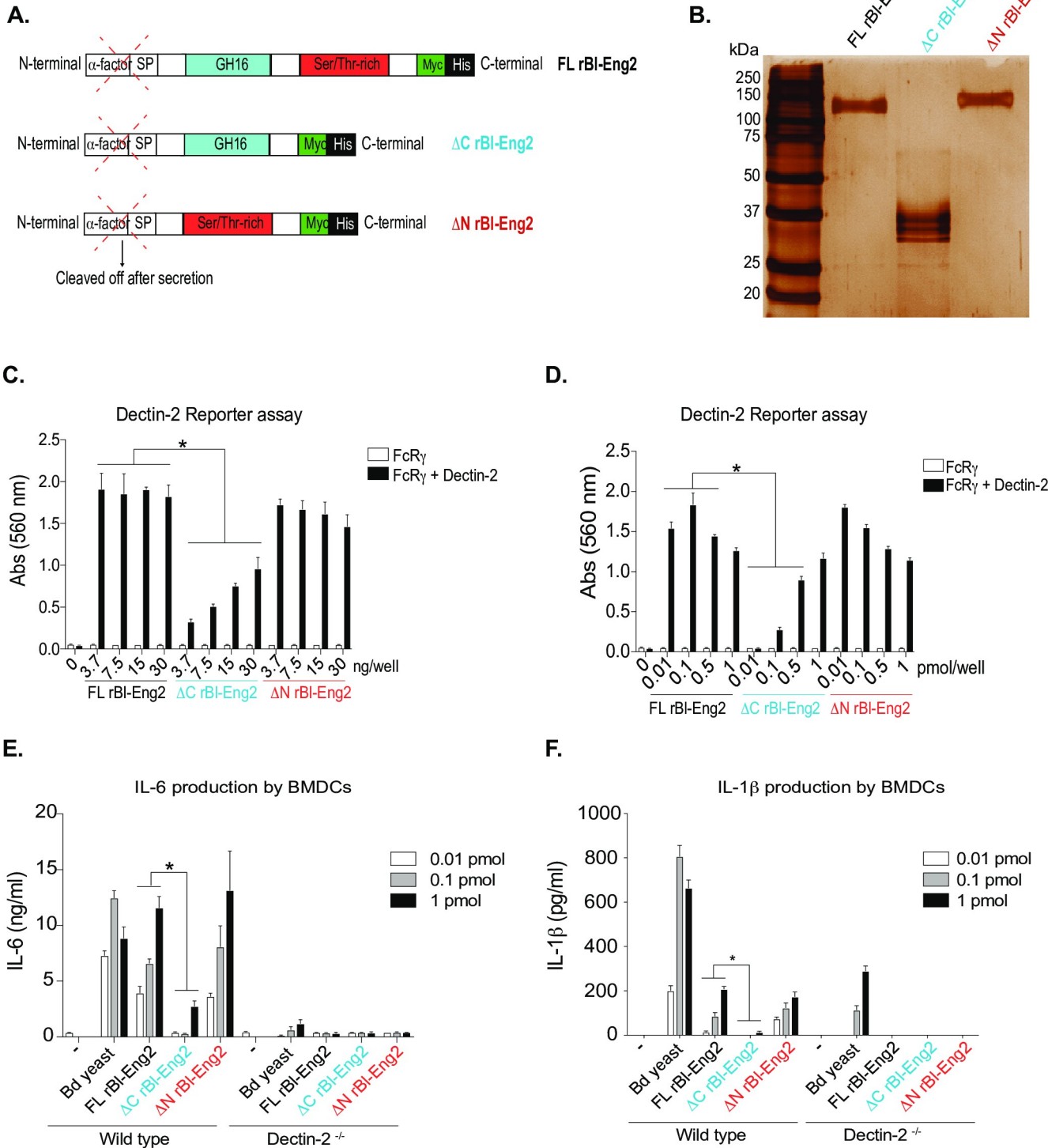

**Fig 3. The C-terminus of Bl-Eng2 stimulates Dectin-2 reporter activity. A)** rBl-Eng2 lacking the C-terminus or the N-terminus was expressed in *P. pastoris*. **B)** Full length and truncated proteins were purified and analyzed by SDS-PAGE. **C+D)** All three proteins were tested in the reporter assay by titrating mass (**C**) or molar amounts (**D**). **E+F)** BMDCs from wild type and Dectin-2[-/-] mice were stimulated with rBl-Eng2 constructs and IL-6 (**E**) and IL-1β (**F**) measured in cell culture supernatants by ELISA. Stimulation with *B. dermatitidis* yeast served as a positive control. *$p < 0.5$ vs. full length Bl-Eng2. The data are representative of three independent experiments.

with long O-linked mannan chains. Conversely, eliminating the N-terminus of the protein did not significantly reduce the molecular weight of the protein.

Whereas eliminating the N-terminus did not reduce Dectin-2 signaling by the reporter cells, removing the C terminus did impair signaling (**Fig 3C and 3D**) when either equivalent mass (**Fig 3C**) or molar (**Fig 3D**) amounts of the proteins were used for the assay. In addition, both ΔN rBl-Eng2 and full length rBl-Eng2 stimulated production of IL-6 and IL-1β by murine bone marrow derived dendritic cells (BMDCs), whereas ΔC rBl-Eng2 triggered sharply reduced responses by the cells (**Fig 3E and 3F**). In summary, these data reveal that Dectin-2 principally recognizes the C terminus and O-linked mannans of Bl-Eng2, which induce pro-inflammatory cytokines that are capable of driving Th17 cell differentiation.

## O-linked glycans of the C-terminus of Bl-Eng2 (ΔN rBl-Eng2) augment T cell expansion, tissue residency, Th1 and Th17 cell differentiation and resistance to *B. dermatitidis* infection

Full length Bl-Eng2 used as an adjuvant in combination with the antigen calnexin, triggered cytokine production by antigen presenting cells, T cell differentiation of corresponding TCR-transgenic 1807 cells and protected mice against pneumonia in a model of lethal pulmonary fungal infection [27]. However, we subsequently discovered that full length Bl-Eng2 also harbors a CD4+ T cell antigen starting at amino acid position 35 in the N terminus of the protein that is immune-dominant and protects against experimental infection [36]. In hindsight, it is unclear whether Bl-Eng2 mediated protection in the former report was attributable to the antigenic or adjuvant properties of Bl-Eng2. To assess whether O-mannans in the C terminus of Bl-Eng2 are sufficient to drive the protein's adjuvant properties *in vivo* we combined ΔN rBl-Eng2 as an adjuvant with two antigens from *B. dermatitidis* for vaccination. First, we formulated the immunodominant T cell epitope of calnexin [37] with ΔN rBl-Eng2 and measured T cell expansion and differentiation of adoptively transferred, congenic (CD90.1+) TCR transgenic 1807 cells following vaccination. The addition of ΔN rBl-Eng2 to the calnexin peptide significantly augmented T cell expansion, Th17 cell differentiation and recruitment to the lung upon challenge with the fungus (**Figs 4A and 3B**). Second, we combined the immuno-dominant T cell epitope of Bl-Eng2 located in the N-terminus of the glycoprotein with ΔN rBl-Eng2 (as an adjuvant) for vaccination and measured the expansion and differentiation of endogenous Bl-Eng2-specific CD4+ T cells with tetramer, and antifungal resistance. The addition of ΔN rBl-Eng2 to the peptide increased T cell expansion, Th1 and Th17 cell differentiation, migration of primed CD4+ T cells into the lung parenchyma and resistance as measured by reduction of lung CFU and weight loss (**Fig 4C–4F**). As a positive control for CD4+ T cell priming, we vaccinated mice with full length Bl-Eng2 since it harbors both the antigen and adjuvant properties [27, 36]. We titrated the amount of peptide used for vaccination. For these experiments, we used equimolar (1x) (0.24 μg) amounts of Bl-Eng2 peptide relative to the amount of peptide present in full length protein. In case the formulation of peptide and adjuvant as separate molecules was less efficient in fostering T cell priming than full length Bl-Eng2 (contains both functional domains in one linear glycoprotein), we used a 10-fold excess (2.4 μg) of peptide. We found that equimolar amounts of peptide in conjunction with ΔN rBl-Eng2 yielded the highest expansion and differentiation of tetramer+ CD4 T cells, and the greatest resistance compared to the 10-fold excess and full length Bl-Eng2 protein (**S5 Fig**). Thus, for subsequent experiments we vaccinated mice with the 1x amount of peptide. In summary, ΔN rBl-Eng2 exhibited bona-fide adjuvant properties in the fungal vaccine model with *B. dermatitidis* augmenting T cell development of antigen-specific CD4+ T cells.

## Calnexin peptide

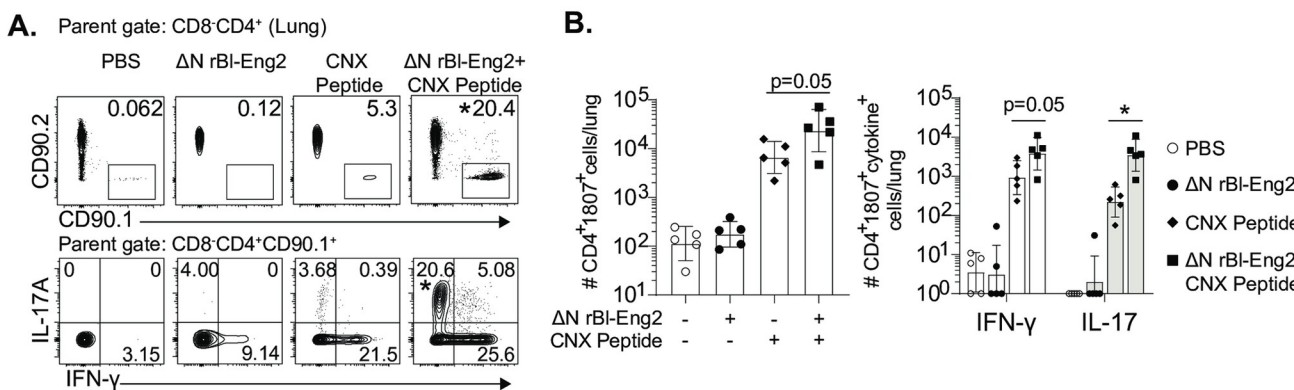

**A.** Parent gate: CD8⁻CD4⁺ (Lung)

PBS · ΔN rBl-Eng2 · CNX Peptide · ΔN rBl-Eng2+ CNX Peptide

**B.**

## Bl-Eng2 peptide

**C.** Parent gate: CD8⁻CD4⁺ (Lung)

**D.**

**E.** Day 14 Post Infection

**F.** Day 14 Post Infection

**Fig 4. The C-terminus of Bl-Eng2 (ΔN rBl-Eng2) augments T cell expansion, tissue residency, Th1 and Th17 cell differentiation and resistance upon vaccination against _B. dermatitidis_ infection.** Two antigens were used to test adjuvant activity: calnexin (CNX) **(A+B),** and Bl-Eng2 peptide **(C-F). (A +B)** Mice received adoptively transferred 1807 T cells prior to vaccination and were subcutaneously vaccinated with 240 ng calnexin peptide and 10μg ΔN rBl-Eng2 (as adjuvant). Two weeks after the boost, mice were challenged with virulent _B. dermatitidis_ yeast. At day 4 post-infection, the frequency (**A**) and number (**B**) of lung 1807 cells (CD90.1+) producing IL-17 and IFN-γ were analyzed by flow cytometry. **C-F)** Mice were vaccinated with Bl-Eng2 peptide and ΔN rBl-Eng2 as above and challenged with _B. dermatitidis_. At day 4 post-infection, the frequency (**C**) and number (**D**) of total tetramer+ T cells in the lungs (top row) are shown, including those in the parenchyma and vasculature (middle row), and those producing IL-17 and IFNγ (bottom row) as analyzed by flow cytometry. Data are from a representative experiment (n = 5 mice/group) of two performed. *$p < 0.5$ vs. the Bl-Eng2 or calnexin peptide alone group. **E)** Lung CFU are displayed as the geometric mean with standard deviation. *$p < 0.05$ vs. Bl-Eng2 peptide group. **F)** Percent body weight change was calculated by weighing the mice at the time of challenge and euthanasia (two weeks post infection). N = 10 mice/group. *$p < 0.05$ vs. Bl-Eng2 peptide group.

## Vaccine adjuvant activity of the C-terminus of Bl-Eng2 (ΔN rBl-Eng2) is glycosylation- and Dectin-2 dependent

Because BMDCs stimulated by ΔN rBl-Eng2 produce IL-6 and IL-1β in a Dectin-2 dependent manner (**Fig 3E and 3F**), we hypothesized that the _in vivo_ adjuvant function of ΔN rBl-Eng2 is also Dectin-2- and glycosylation-dependent. To test our hypothesis, we vaccinated Dectin-2-/- mice with ΔN rBl-Eng2 and wild type mice with deglycosylated ΔN rBl-Eng2 (**S6 Fig**) and analyzed the development of tetramer+ T cells. In the absence of either Dectin-2 or glycosylation, tetramer+ T cells failed to expand (**Fig 5A and 5B**), produce IL-17 and IFN-γ (**Fig 5C, 5D and 5F**) or protect against fungal infection (**Fig 5E**). Thus, the vaccine adjuvant properties of ΔN rBl-Eng2 require O-mannosylation triggering of Dectin-2 signaling.

## ΔN rBl-Eng2 augments resistance against disseminated infection with _C. albicans_ and pulmonary infection with _C. neoformans_

To investigate whether ΔN rBl-Eng2 could be harnessed as a vaccine adjuvant against other fungal pathogens, we chose to test adjuvant efficacy in a disseminated model of _Candida albicans_ infection and a respiratory model of _Cryptococcus neoformans_ infection. For candidiasis, we vaccinated C57BL/6 mice with the Als3 peptide (**Fig 6A**) since formulation with curdlan protects vaccinated mice against infection [38]. The inclusion of ΔN rBl-Eng2 to vaccination with Als3 peptide increased the number of tetramer+ CD4+ T cells in the spleens (**Fig 6B**), augmented the differentiation of Th17 cells in the skin draining lymph nodes (**Fig 6C**), reduced the fungal burden in the kidneys (**Fig 6D**) and increased survival (**Fig 6E**) compared to peptide vaccinated and naïve control mice. To test ΔN rBl-Eng2 adjuvant in the pulmonary model of _C. neoformans_ infection, we vaccinated BALB/c mice with the Cda2 protein (**Fig 6F**) [39]. The addition of ΔN rBl-Eng2 reduced lung CFU compared to peptide vaccinated and naïve control mice (**Fig 6G**). In summary, ΔN rBl-Eng2 adjuvant significantly augmented vaccine-induced resistance against additional pulmonary and systemic mycoses.

## Anti-viral vaccination with ΔN rBl-Eng2 as adjuvant augments T cell expansion, differentiation and tissue residency of anti-viral CD8+ Tc17 cells and protection against IAV infection

Vaccine-induced resistance to respiratory IAV is principally mediated by nucleoprotein (NP) specific CD8+ T cells when mice are vaccinated with NP and Adjuplex by the mucosal (intranasal) route [40]. We sought to investigate whether ΔN rBl-Eng2 augments mucosal immunity to IAV infection (**Fig 7A**). The addition of ΔN rBl-Eng2 to vaccination with NP and adjuplex increased the number of tetramer+ NP366-specific CD8+ T cells that migrated to the lung (**Fig 7B**) and lung parenchyma (**Fig 7C**) upon IAV infection,. The frequency of NP366-specific CD8+ T cells that expressed CD69 and CD103 was increased in the parenchyma vs. the

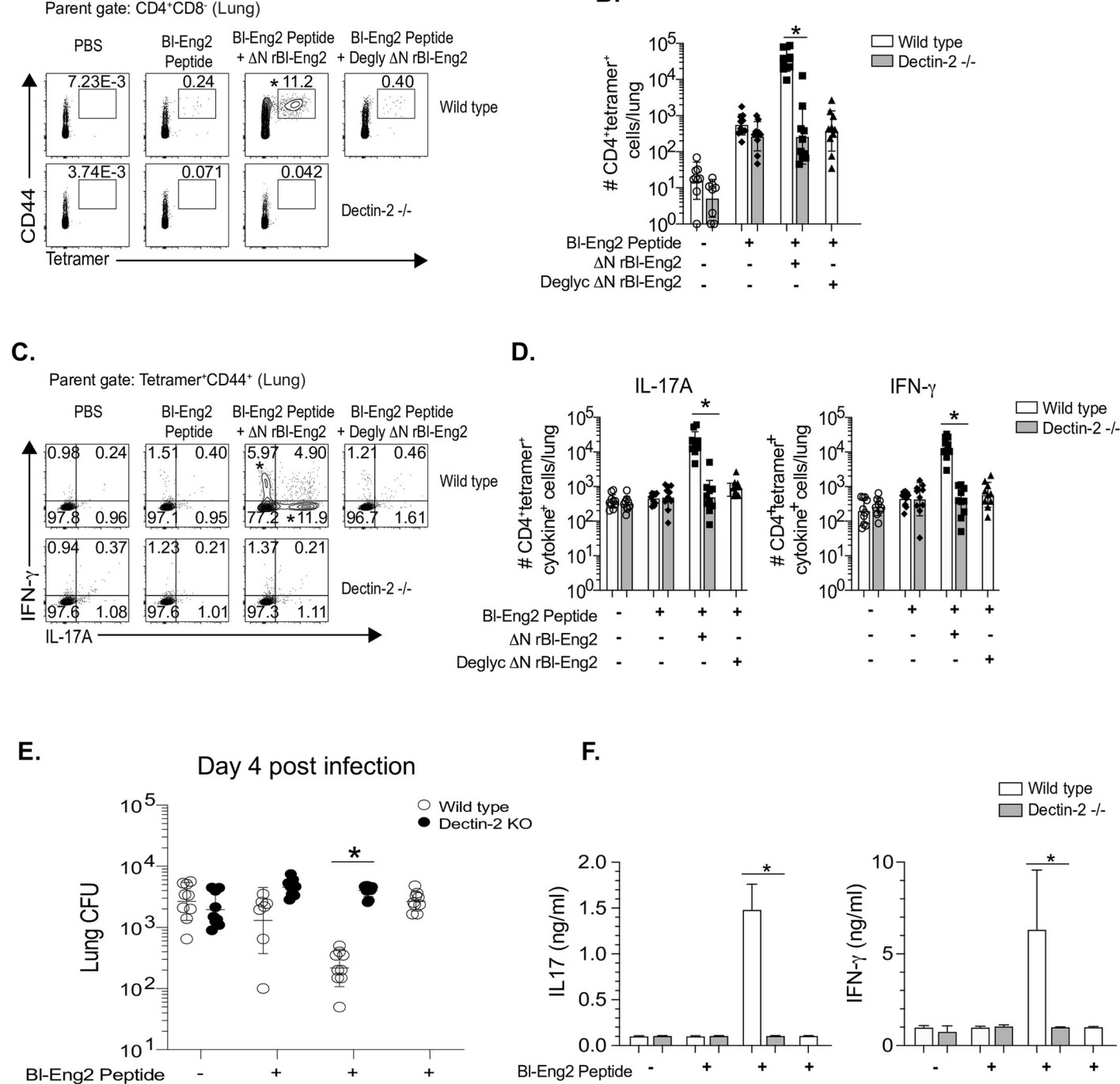

**Fig 5. Adjuvant activity of the C-terminus of Bl-Eng2 (ΔN rBl-Eng2) is glycosylation- and Dectin-2 dependent.** Wild type mice were vaccinated with deglycosylated ΔN rBl-Eng2 and Dectin-2⁻/⁻ mice received untreated ΔN rBl-Eng2. Percentage (**A**) and number (**B**) of tetramer⁺ lung T cells were analyzed 4 days post-infection. Percentage (**C**) and number (**D**) of IL-17A and IFN-γ producing Bl-Eng2-specific CD4⁺ T cells. Data represent an average of two independent experiments (n = 10 mice/group). *$p<0.05$ for WT vs KO vaccine groups. **E)** Lung CFU are displayed as geometric mean with standard deviation. *$p<0.05$ for WT vs KO vaccine groups. **F)** IFNγ and IL-17A from cell culture supernatants of Bl-Eng2 peptide-stimulated splenocytes as measured by ELISA. *$p<0.05$ for vaccinated WT vs. KO groups.

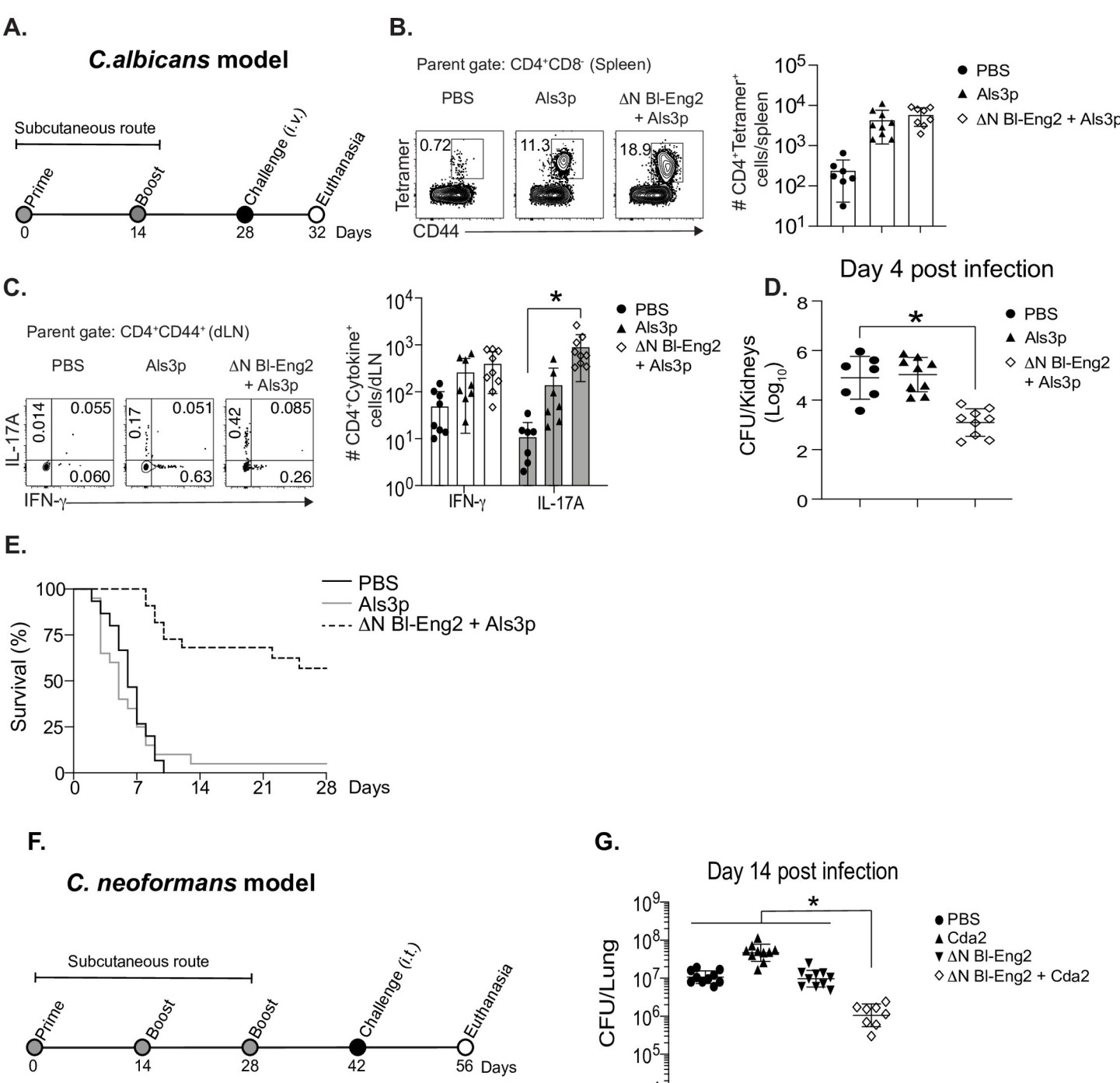

**Fig 6. Vaccination with ΔN rBl-Eng2 augments protective immunity against systemic candidiasis and pulmonary cryptococcosis. A)** Mice were subcutaneously vaccinated twice, two weeks apart with ΔN rBl-Eng2 and Als3 peptide and challenged with *C. albicans* strain ATCC SC5314 (intravenously). **B)** The frequency and number of Als3-specific CD4+ T cells in the spleen at day 4 post-challenge after tetramer pull down. **C)** The percentage and number of IL-17A and IFN-γ producing Als3-specific CD4+ T cells in the draining lymph nodes at day 4 post-infection. **D)** *C. albicans* CFU in the kidneys at day 4 post-challenge and **E)** survival curve. **F)** Mice were SC vaccinated three times, two weeks apart and challenged with *C. neoformans* var *grubii* strain KN99 (intracheally) two weeks after the last boost. **G)** *C. neoformans* lung CFU at day 14 post-infection. Data are the average of two indendepent experiments (n = 10 mice/group for cellular analysis and CFU. n = 20 mice/group for survival). $^*p < 0.05$ for ΔN rBl-Eng2 + Als3 or Cda2 vs PBS.

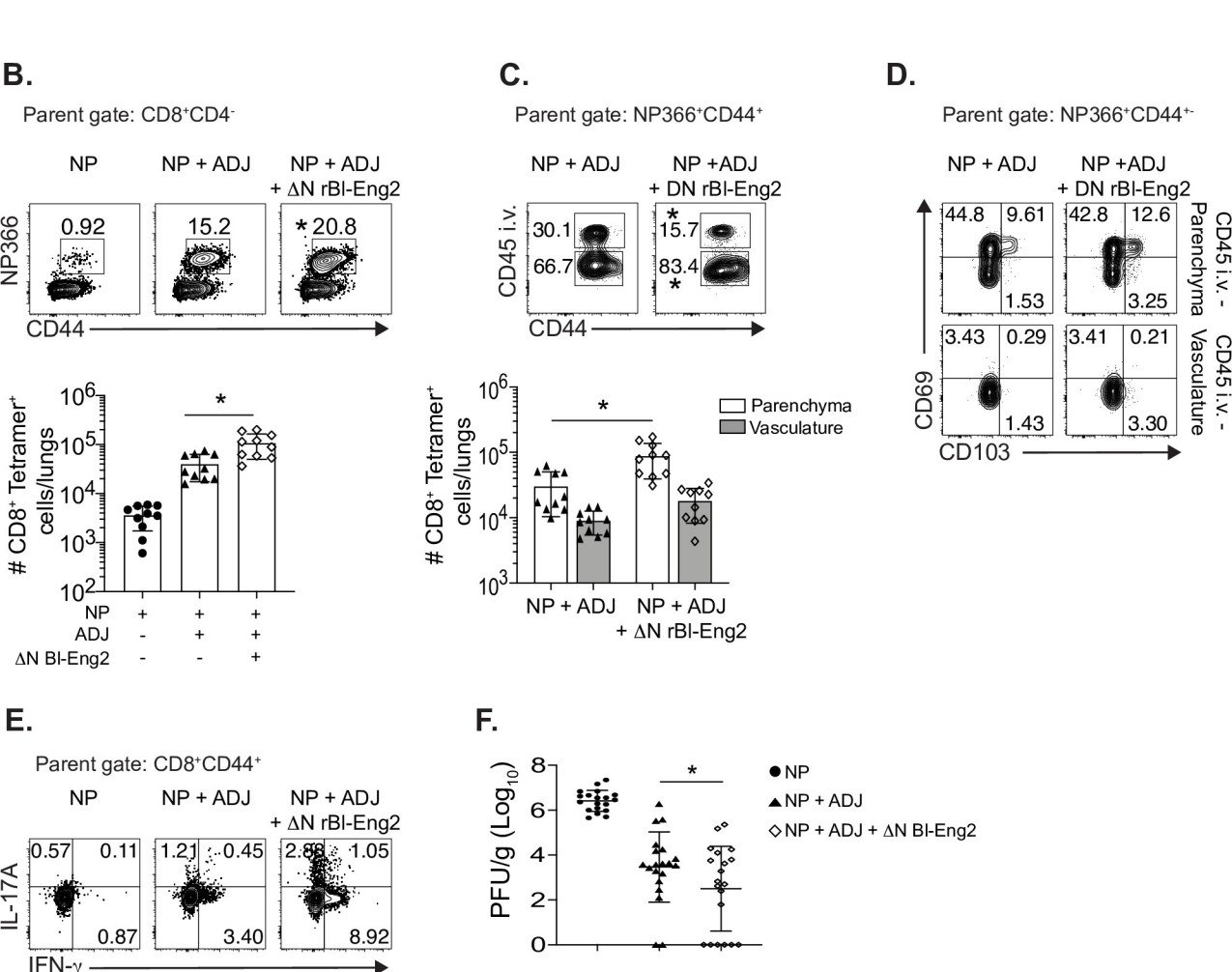

**Fig 7. The C-terminus of Bl-Eng2 (ΔN rBl-Eng2) serves as an adjuvant, augmenting influenza specific CD8⁺ T cell expansion, tissue residency, Th17 and Tc17 cell differentiation and resistance to IAV infection. A)** Mice were intranasally vaccinated with nucleoprotein (NP), adjuplex and/or ΔN rBl-Eng2 protein and challenged with influenza A virus (IAV) (strain PR8). **B)** CD8⁺ NP366-specific tetramer⁺ T cells were enumerated in the lungs day 6 post infection. **C)** Anti-CD45 mAb was injected i.v 3 minutes before euthanizing mice to distinguish and enumerate parenchymal (CD45⁻) from vasculature associated lung cells (CD45⁺). **D)** CD69 and CD103 expression by parenchymal (CD45⁻) vs. vascular (CD45⁺) NP366-specific CD8⁺ T cells. **E)** IL-17A and IFN-γ production by peptide-stimulated lung CD8⁺ T cells. **F)** Viral load in the left lung was assessed by plaque assay. Data are the average of two independent experiments performed (n = 20 mice/group). Male and female cohorts of mice were used for these experiments and no gender difference was observed. $^*p<0.05$ for adjuplex + NP vs Adjuplex + NP + ΔN rBl-Eng2.

vasculature (**Fig 7D**) indicating that most of the tetramer⁺ CD8⁺ T cells in the parenchyma are likely $T_{RM}$ cells. We also found a trend that the addition of ΔN rBl-Eng2 to vaccination with NP and adjuplex increased the frequencies of IL-17 and IFN-γ producing tetramer⁺ NP366-specific CD8⁺ T cells (**Fig 7E**). Importantly, the addition of ΔN rBl-Eng2 to adjuplex reduced lung viral burden (plaque forming units (PFU) when compared to mice vaccinated with adjuplex (**Fig 7F**). In summary, mucosal vaccination with ΔN rBl-Eng2 augmented the

development of NP366-specific Tc1 and Tc17 cells, tissue residency in the lung parenchyma and resistance to IAV infection.

## Discussion

The hallmark of an effective adjuvant is the ability to elicit the type of adaptive immune response that mediates protective immunity to specific pathogens. We have observed that protective immunity to inhaled fungi is best induced by subcutaneous vaccination and mediated by CD4$^+$ T cells that migrate from the secondary lymphoid organs to the lung upon infection and produce type 1 and 17 cytokines [13, 15], whereas lung tissue resident memory T cells (T$_{RM}$) induced by mucosal vaccination do not protect [36]. In contrast, host immunity to respiratory viral infection with Influenza A is most efficiently induced at the respiratory mucosa, but not via the subcutaneous route and requires Tc1 and Tc17 cells in the airways and lung parenchyma [40–42].

In this manuscript, we have elucidated how the newly described Dectin-2 ligand Bl-Eng2 acts as an adjuvant and regulates the development of antigen-specific T cell responses and vaccine immunity to inhaled fungi and virus. We found that triggering the signaling of Dectin-2 during vaccination, augments the expansion, differentiation, and tissue residency of antigen-specific CD4$^+$ and CD8$^+$ T cells in the lung upon fungal and viral challenge and protects mice against experimental infection. In the vaccine model for *B. dermatitidis*, the addition of ΔN rBl-Eng2 (as an adjuvant) to calnexin or Bl-Eng2 peptide greatly augmented the expansion and recruitment of antigen-specific T cells to the lung on recall. Importantly, the addition of ΔN Bl-Eng2 as adjuvant augmented the differentiation of antigen-specific CD4$^+$ T cells into Th1 and especially Th17 cells, which are tightly correlated with vaccine resistance to fungi [13]. We recently reported that primed T cells from subcutaneously vaccinated mice migrate swiftly from the secondary lymphoid organs via the vasculature to the lung parenchyma to mediate vaccine-induced protection [36]. We demonstrate here that addition of ΔN Bl-Eng2 as adjuvant during vaccination fosters efficient migration of primed tetramer$^+$ T cells into the lung parenchyma. Thus, subcutaneous delivery of ΔN Bl-Eng2 as an adjuvant in the *Blastomyces* vaccine model fosters protective immunity in association with augmented expansion and differentiation of antigen-specific Th1 and Th17 cells and their efficient migration into the lung parenchyma upon fungal infection. Vaccination against pulmonary infection with *C. neoformans* and systemic infection with *C. albicans* provided proof of concept that ΔN rBl-Eng2 can be exploited as an adjuvant against multiple fungi that invade the host by respiratory and systemic routes.

Engagement of Dectin-2 signaling by ΔN Bl-Eng2 adjuvant also augmented mucosal immunity mediated by NP-specific Tc1 and Tc17 cells. The combination of adjuplex with ΔN Bl-Eng2 yielded additive adjuvant effects. Adjuplex mediates its adjuvant effects by efficiently delivering protein antigens into antigen presenting cells and cross-priming CD8$^+$ CTLs [40]. By mixing ΔN Bl-Eng2 with adjuplex we augmented the expansion and recruitment of NP366-specific CD8$^+$ T cells into the lung parenchyma. The establishment of primed NP-specific T$_{RM}$ cells in the lung parenchyma was previously found to be critical for NP-vaccine induced protection [40]. In the current study, the addition of ΔN Bl-Eng2 to adjuplex increased tissue residency of NP366$^+$ CD8$^+$ T cells as evidenced by their surface expression of CD69 and CD103. The addition of ΔN Bl-Eng2 to adjuplex induced NP-specific CD8$^+$ T cells to become polarized into Tc1 and Tc17 cells, which have been shown to protect against influenza A infection [43]. Most importantly, the addition of ΔN Bl-Eng2 to adjuplex significantly augmented protection against IAV infection. In summary, the addition of ΔN Bl-Eng2 as an adjuvant augmented vaccine induced resistance by the subcutaneous (systemic) and mucosal

routes against respiratory and systemic/disseminated infections with both fungal and viral pathogens. The adjuvant was also able to augment resistance mediated by both CD4[+] and CD8[+] T cells.

Dectin-2 recognizes fungal high mannose structures on their surfaces [21, 24]. Among the known Dectin-2 ligands, including furfurman from *Malassezia* spp, Man-LAM from *M. tuberculosis* and MP98 from *Cryptococcus neoformans*, Bl-Eng2 is the superior ligand based on its ability to elicit cytokine responses through Dectin-2 signaling [27]. Thus, using the most potent ligand for Dectin-2 recognition and engagement allowed us to unravel the molecular domain requirements for receptor signaling and adjuvancy. Here, we identified 4 N-linked glycans that attached to asparagine residues via N-acetylglucosamine residues within the N-terminus of Bl-Eng2. Enzymatic removal of N-glycans did not alter Dectin-2 recognition of Bl-Eng2. We cannot rule out the possibility that N-glycans are recognized by Dectin-2 since the large number of remaining O-glycans could have masked a contribution of N-glycans to Dectin-2 recognition and signaling. Most importantly, through biochemical and molecular analyses we determined that the O-linked α1,2 mannose chains located at the C-terminus of Bl-Eng2 are the principal structures recognized by Dectin-2. The O-mannose residues are attached to serine/threonine amino acids and range in length between 2–19 residues. Our findings reinforce a previous report in which NMR and methylation analysis of a mucin-like serine-rich glycoprotein from *Malassezia* revealed an α-1,2-linked mannobiose recognized by Dectin-2 [29].

In summary, we have analyzed and pinpointed the structural basis for Bl-Eng2 engagement of Dectin-2 during vaccine priming, and demonstrated that these interactions augment the development of antigen-specific CD4[+] and CD8[+] T cell responses with phenotypic, functional, and localization features designed to promote resistance against inhaled fungal and viral infections at the respiratory mucosa. What is more, Bl-Eng2 promoted an adjuvant effect upon either systemic (subcutaneous) or mucosal vaccination and catalyzed the generation of both migratory and tissue resident T cells in the lung.

## Methods

### Ethics statement

The animal studies performed were governed by protocols M005891 as approved by the IACUC committees of the University of Wisconsin-Madison Medical School. Animal studies were compliant with all applicable provisions established by the Animal Welfare Act and the Public Health Services (PHS) Policy on the Humane Care and Use of Laboratory Animals.

### Fungi

Wild-type, virulent *B. dermatitidis* ATCC strain 26199 was used for this study and grown as yeast on Middlebrook 7H10 agar with oleic acid-albumin complex (Sigma) at 39˚C. Virulent *C. albicans* yeast of ATCC strain SC5314 were used in some experiments and grown as yeast on YPD agar or broth at 30˚C. *C. neoformans* var. *grubii* strain KN99 were maintained as glycerol stocks -80C and initially cultured on YPD agar at 30˚C.

### Mouse strains

Inbred wild type C57BL/6 mice were obtained from Jackson Laboratories and Dectin-2[-/-] mice [32] were bred at the University of Wisconsin in Madison. Balb/c mice were used for the *C. neoformans* vaccine model at the University of Massachussetts. *Blastomyces*-specific TCR Tg 1807 mice were generated in our laboratory and were backcrossed to congenic Thy1.1[+] mice

as described [44]. Male and female mice were 7–8 weeks old at the time of these experiments. Mice were housed and cared for as per guidelines of the University of Wisconsin Animal Care Committee who approved all aspects of this work.

## Generation and expression of intact Bl-Eng2, and derivatives with C-terminal (ΔC) and N-terminal (ΔN) deletions

Full length Bl-Eng2 was expressed in *Pichia pastoris* and purified as described [27]. To create ΔC and ΔN Bl-Eng2, we synthesized codon-optimized truncated open reading frames as gBlocks (Integrated DNA Technologies) and inserted them in-frame into the *Xho*I/*Xba*I sites of the pPICZαA vector (Invitrogen) by the Gibson Assembly reaction (New England Biolabs). The resulting expression vectors were confirmed for correct fusion protein open reading frames by sequencing and each was linearized with *Pme*I (New England Biolabs) and transformed into the *P. pastoris* strain X-33 (Invitrogen) by electroporation. The *Pichia* transformants were grown to high cell density in BMGY (buffered-glycerol complex medium) and recombinant protein expression was induced by subsequent growth in BMMY (Buffered-Methanol complex medium) broth at 30˚C until maximal growth and target protein expression was induced by the addition of 0.1% methanol every 6–8 hours for 72 hours. The secreted protein in the culture supernatant was purified using Ni-NTA agarose (Qiagen) according to the manufacturer's protocol and dialyzed against PBS. The purity of recombinant Bl-Eng2 was assessed by SDS-PAGE and western blot using an anti-His antibody (Cell signaling Technology).

## Cloning and expression of Bl-Eng2 in *E. coli*

A synthetic DNA fragment (Integrated DNA Technologies) containing a full length, codon-optimized Bl-Eng2 open reading frame with an upstream enterokinase site and downstream c-Myc and 6X histidine tags was cloned into the SpeI and XhoI sites of the pET43.1b plasmid (Novagen); this forms a fusion protein with an upstream NusA tag, which enhances intracellular solubility. The resulting plasmid was transformed into the *E. coli* BL21(DE3) strain and protein expression was induced in Overnight Express™ Instant TB Medium (Novagen) at 37˚C. The cells were lysed using a French Press and the recombinant protein was purified using nickel affinity chromatography, as with protein produced in *Pichia* above. Western blot analysis confirmed that the higher molecular band (boxed in red) in Fig 1A contained recombinant Bl-Eng2.

## Deglycosylation of Bl-Eng2

Deglycosylation of Bl-Eng2 was performed by using the GlycoProfile™ IV Chemical Deglycosylation kit (Sigma-Aldrich) according to the manufacturer's specifications. Treatment with trifluormethanosulfonic acid (TFMS) was used to remove *N*- and *O*-glycans from Bl-Eng2 while preserving the protein backbone [34]. PNGase F (New England Biolabs) digestion was employed to remove N-linked oligosaccharides from Bl-Eng2, and Proteinase K (Promega) was used to digest the protein back-bone. Pro-Q™ Emerald 300 glycoprotein gel stain kit (Invitrogen) was used to detect the presence of sugar in Bl-Eng2 before and after deglycosylation. Colloidal Coomassie blue staining (Invitrogen) or silver stain was used to evaluate protein homogeneity and migration in SDS-PAGE.

## Dectin-2 reporter assay

B3Z reporter cells expressing Dectin-2 have been described [20]. About $10^5$ B3Z cells per well in a 96-well plate were incubated for 18 hr with plate-coated ligands. β-galactosidase (lacZ)

activity was measured in total cell lysates using CPRG (Roche) as a substrate. $OD_{560}$ was measured using $OD_{620}$ as a reference. 2B4-NFAT-GFP reporter cells expressing Dectin-2 and a Dectin-2$^{QPD}$ mutant (E168Q/N170D) were used as described [31].

## Glycosylation analysis

In this work, the *Pichia pastoris* X33 expressed Bl-Eng2 protein was digested with multiple proteases including Trypsin, Glu-C, and Chymotrypsin to obtain a more comprehensive glycopeptide coverage. Briefly, the sample was dissolved in PBS (pH 7.4) and boiled in 95˚C waterbath for 10 min and then treated with dithiothreitol and iodoacetamide for disulfide bond reduction and alkylation, respectively. The samples were protease-digested at 37˚C overnight and desalted with C-18 zip-tip. The sample was reconstituted in 0.1% formic acid (FA) and loaded onto a Fusion Lumos Orbitrap instrument for LC-MS/MS analyses with Electron Transfer/Higher-Energy Collision Dissociation (EThcD) fragmentation.

Glycopeptides were identified based on searching the MS data using the Byonic software. Mass tolerance of ±10 ppm was used for precursor ions and ±0.03 Da for fragmentation ions, and EThcD fragmentation type was selected. Oxidation of methionine and deamidation of asparagine and glutamine were specified as rare variable modifications (rare), while carbamidomethylation of cysteine was a set as a fixed modification. O-linked glycosylation of serine and threonine and N-linked glycosylation of asparagine were separately set as a common variable modification and glycan database was added manually which includes O-linked glycans hex(1–30), hex(1–30)phospho and N-linked glycans HexNAc(2)Hex(1–30), and HexNAc(2) Hex(1–30)phospho. A total of 1 common and 2 rare modifications were allowed per identification. Trypsin and GluC specificity was used with up to 5 missed cleavages allowed and Chymotrypsin specificity allowing for up to 9 missed cleavages was used. The acquired Byonic data were filter at 1% false discovery rate (FDR) at the peptide spectral match level using the 2D-FDR score and removed glycopeptide identifications that had a Byonic score below 150, Delta Mod. score below 10, |logProb| value below 1, and PEP2D above 0.05, and all the identified results were confirmed with manual inspection.

Linkage analysis was performed by treating 100 µg of Bl-Eng2 protein with anhydrous hydrazine and incubated at 95˚C for 6 hours, a condition that customarily removes N and O-glycans from the proteins. Hydrazine was then removed by drying with N2 gas. A solution of acetic anhydride in sodium bicarbonate was added to the dried sample and incubated for 40 minutes for N-acetylation followed by mild acid hydrolysis of the sample by Cu (II) acetate. The reaction mixture was passed through Dowex (H+ form) for desalting and the flow through collected. The sample was passed through a C18 Sep-pak cartridge, and the glycans eluted with 5% acetic acid and dried. The dried sample was permethylated using NaOH/DMSO base and Iodomethane (MeI). For determination of sugar linkages, partially methylated alditol acetates were prepared from permethylated glycans as described [45]. Briefly, permethylated glycans were hydrolyzed with 2M TFA at 100˚C for 4 h, followed by reduction with 1% NaBH4 in 30mM NaOH and acetylation with acetic anhydride/pyridine (1:1, v/v) at 100˚C for 15 min [46]. The partially methylated alditol acetates obtained were analyzed by GC-MS (Agilent 7890A GC interfaced to a 5975C MSD).

## Generation of murine bone-marrow dendritic cells (BMDC)

Bone marrow-derived dendritic cells (BMDCs) were obtained from the femurs and tibias of wild type and Dectin-2$^{-/-}$ mice. Each bone was flushed with 10 ml of 1% FBS in RPMI through a 22G needle. Red blood cells were lysed with ACK buffer and the cells were resuspended in 10% FBS in RPMI medium. In a petri dish, $2 \times 10^6$ bone marrow cells were plated in 10 ml of

RPMI containing 10% FBS plus 100 μg/ml penicillin-streptomycin (HyClone), 2-mercaptoethanol and 20 ng/ml of rGM-CSF. The culture media were refreshed every three days and BMDCs were harvested after 10 days for *in vitro* assays.

## Cytokine production by murine BMDCs

$2x10^5$ BMDCs/well in 96-well plate were incubated with equimolar amounts of full-length, ΔC- or ΔN-Bl-Eng2. 24 hours later, the plate was spun down, cell culture supernatants carefully collected and IL-1β and IL-6 levels measured by ELISA (R&D systems) according to the manufacturer's specifications.

## *In vivo* adjuvant testing in a fungal and influenza A virus vaccine models

To test the adjuvant function for vaccination *in vivo* we used three fungal models of infection and one viral model: 1) *B. dermatitidis*: we formulated ΔN-Bl-Eng2 with either Calnexin or Bl-Eng2 peptide. Mice were subcutaneously vaccinated twice, 2 weeks apart, with either 0.24 or 2.4 μg of Bl-Eng2 [36] or calnexin peptide [37] and 10 μg ΔN-Bl-Eng2 emulsified in incomplete Freund's adjuvant (IFA). Prior to the vaccination with calnexin peptide, mice received 2 x $10^5$ adoptively transferred naïve 1807 TCR transgenic T cells [44]. As controls, animals were vaccinated with only PBS, peptide (antigen), ΔN-Bl-Eng2 (adjuvant), deglycosylated ΔN-Bl-Eng2 (adjuvant), or full-length Bl-Eng2 (adjuvant). Two weeks after boost, mice were intratracheally challenged with 2 x $10^4$ virulent *B. dermatitidis* yeasts (ATCC 26199) and analyzed for lung T cell responses at day 4 post-infection, and fungal lung CFU at day 4 and 2 weeks post infection. Lung CFU were plated on brain heart infusion agar plates with 100 μg/ml penicillin and streptomycin.

2) *C. albicans*: We formulated ΔN-Bl-Eng2 with Als3 peptide as described for vaccination with curdlan [47]. Mice were vaccinated with 2 μg of Als3 peptide and 10 μg ΔN-Bl-Eng2 SC twice, two weeks apart and intravenously challenged with $2x10^5$ *C. albicans* yeast of strain ATCC SC5314 two weeks after the boost. At day 4 post-infection, mice were euthanized and the number of Als3-specific CD4$^+$ T cells in the spleen determined following tetramer pull-down using magnetic columns (Miltenyi Biotech). Fungal burden and cytokine production were evaluated in the kidneys and skin draining (brachial) lymph nodes, respectively. Kidney homogenates were plated on YPD agar plates with 100 μg/ml penicillin and streptomycin to quantify *C. albicans* CFU. Another set of mice were evaluated for survival.

3) *C. neoformans*: mice were vaccinated with 10 μg Cda2 protein and 10 μg ΔN-Bl-Eng2 twice, two weeks apart. Two weeks after the boost, the mice were intratracheally challenged with 2 x $10^4$ *C. neoformans* var. *grubii* strain KN99 and the lung fungal CFU determined two weeks post-infection [39]. *Cryptococcus* lung CFU were plated on Sabouraud dextrose agar plates with 100 μg/ml penicillin and streptomycin.

**Viral model.** Mice were intranasally vaccinated twice, three weeks apart, with 10 μg Influenza A H1N1 nucleoprotein (Sino Biological Inc., 11675-V08B) alone, together with 5% Adjuplex (Empirion, LLC, Columbus OH), or with 5% Adjuplex and 10 μg ΔN-Bl-Eng2 together. Mice were challenged with $10^4$ pfu of Influenza A virus strain A/PR/8/34 H1N1 (PR8) 40 days after the boost. Cellular response and viral load (PFU) in the lungs were evaluated at day 6 post infection.

## Intravascular staining, T cell stimulation and flow cytometry

Mice were injected intravenously with 2 μg fluorochrome labeled anti-CD45, and 3 minutes later the lungs were harvested. The lungs were dissociated in Miltenyi MACS tubes and digested with collagenase D (1mg/ml) or collagenase B (2mg/ml, for Influenza) and DNase

(1μg/ml) for 25 minutes at 37˚C. The digested lungs were resuspended in 5ml of 40% percoll (GE healthcare, cat 17-0891-01); 3 ml of 66% percoll was underlaid. Samples were spun for 20 minutes at 2,000 rpm at room temperature. The lymphocytes layer was collected and resuspended in complete RPMI (10% FBS, and 100 μg/ml Penicillin/Streptomycin). For the influenza vaccine model, the percoll separation was not performed. Instead after digestion of the lungs, the red blood cells were lysed with ACK buffer, the samples filtered in 40 μM cell strainers and the cells resuspended in complete RPMI. For *ex vivo* stimulation, lung T cells were incubated at 37˚C for 5 hours with 5 μM Bl-Eng2 or calnexin peptide and 1μg anti-mouse CD28 (BD, cat 553294). After 1h, GolgiStop™ (BD, cat 554724) was added to each well. For Influenza experiments, the cells were stimulated with 1.2 μM peptide (NP366), GolgiPlug and IL-2. All FACS samples were stained with LIVE/DEAD™ Fixable Near-IR Dead Cell Stain (Invitrogen) and Fc Block (BD) for 10 minutes at room temperature. T cells were stained with Bl-Eng2 and Als3p tetramer for 1 hour at room temperature or with N366 tetramer for 1.5 hour at 4˚C. Then, the cells were stained for surface antigens and intracellular cytokines. All panels included a dump channel (B220, CD11b, CD11c and NK1.1). Gatting strategy is represented in S4 Fig. 50 μl of AccuCheck Counting Beads (Invitrogen, cat PCB100) was added to the samples to determine the absolute cell count. Flow samples were collected at the University of Wisconsin Carbone Center Flow Lab on a BD LSR Fortessa that was purchased with the NIH shared instrumentation grant 1S100OD018202-01.

**Tetramer enrichment for Als3-specific T cells.** Spleens were harvested at day 4 postchallenge, carefully dissociated and the red blood cells lysed with ACK buffer for 3 minutes. The cell pellet was washed with sorter buffer (PBS + 1% FBS) and the cells stained with Als3 MHC class II tetramer-PE (from the NIH tetramer core facility at Emory, Atlanta, GA) for 1 hour at room temperature. Tetramer-positive T cells were subjected to a pull down [37] using Miltenyi LS columns and enriched cells were surface-stained for Flow Cytometry.

**Surface panel cocktail for calnexin peptide experiments.** CD90.1 BB515 (BD, clone OX-7, cat 564607), CD90.2 BUV395 (BD, clone 53-2-1, cat 565257), CD4 BUV737 (BD, clone RM4-5, cat 565246), CD8 PerCP-Cy5.5 (Biolegend, clone 53–6.7, cat 100734), CD44 BV650 (Biolegend, clone IM7, cat 1033049), CD11b APC (Biolegend, clone M1/70, cat 101212), CD11c APC (Biolegend, clone N418, cat 117310), NK1.1 APC (Biolegend, clone PK136, cat 108710), B220 APC (Biolegend, clone RA3-62B, cat 103212).

**Surface panel cocktail for Bl-Eng2 peptide experiments.** CD45 AF488 (Biolegend, clone 30-F11, cat 103122), MHC class II tetramer-PE (from the NIH tetramer core facility at Emory, Atlanta, GA), CD4 BUV395 (BD, clone GK1.1, cat 563790), CD8 PerCP-Cy5.5 (Biolegend, clone 53–6.7, cat 100734), CD44 BV650 (Biolegend, clone IM7, cat 1033049), CD11b APC (Biolegend, clone M1/70, cat 101212), CD11c APC (Biolegend, clone N418, cat 117310), NK1.1 APC (Biolegend, clone PK136, cat 108710), B220 APC (Biolegend, clone RA3-62B, cat 103212).

**Surface panel cocktail for Influenza.** CD45 PE, MHC class I NP366 tetramer-APC (from the NIH tetramer core facility at Emory, Atlanta, GA), CD103 FITC, CD127 PerCP-Cy5.5, CD62L PE-CF594, CD69 PE-Cy7, CD44 BV510, CD49a BV605, KLRG1 BV711, CX3CR1 BV786, PD-1 APC-R700, CD8 BUV 395, CD4 BUV496, CXCR3 BUV805 (BD, clone CXCR3-173, cat 748700), CD11b APC-Cy7 (clone M1/70, cat 117324), CD11c APC-Cy7 (Biolegend, clone N418, cat 101226), B220 APC-Cy7 (Biolegend, clone RA3-6B2, cat 103224), NK1.1 APC-Cy7 (Biolegend, clone PK136, 108724).

For intracellular staining (Bl-Eng2, Als3 and calnexin): IL-17 PE and IFN-g PE-Cy7. For Influenza the following antibodies were used: IL-17 FITC, IL-22 PerCP-Cy5.5, IL-2 PE-CF594, GM-CSF PE-Cy7, TNF-a BV421, IFN-γ APC.

**Surface panel cocktail for Als3 peptide experiments.** MHC class II tetramer-PE (from the NIH tetramer core facility at Emory, Atlanta, GA), CD4 BUV737 (BD, clone GK1.1), CD8

PerCP-Cy5.5 (Biolegend, clone 53–6.7, cat 100734), CD44 BV785 (Biolegend, clone IM7), CD11b APC (Biolegend, clone M1/70, cat 101212), CD11c APC (Biolegend, clone N418, cat 117310), NK1.1 APC (Biolegend, clone PK136, cat 108710), B220 APC (Biolegend, clone RA3-62B, cat 103212).

### *Ex vivo* stimulation of splenocytes from vaccinated mice

Splenocytes from vaccinated mice were harvested at day 4 post challenge, resuspended in complete RPMI and plated in a 96-well plate ($1x10^6$ cells/well) containing 20 μg/ml Bl-Eng2 protein or peptide. 4 days later, the cell culture supernatants were collected and IL-17A and IFN-γ levels measured by ELISA (R&D systems) according to manufacturer specifications.

### Statistics

Statistical analysis was performed in Prism (GraphPad). A one-way ANOVA with Turkey's test or an unpaired two-tailed t test was used for multiple comparison and between two groups, respectively. For comparison of cell count and CFU, the data were log transformed before the statistical test. Survival curves were analyzed by Mantel-Cox test. A $p \leq 0.05$ was considered statistically significant.

### Supporting information

**S1 Fig. The Ser/Thr-rich region of Bl-Eng2 harbors 68 *in silico* predicted O-glycosylation sites. A)** Schematic of rBl-Eng2 domain structure. The protein consists of the catalytic glycosyl hydrolases family 16 domain (GH16, blue) and a Serine/Threonine-rich region (red). The alpha factor signal promotes expression in *Pichia* and the Myc tag (red) and 6x Histidine tag (black) enable purification. (**B**) Amino acid sequence of the Ser/Thr-rich region predicts the presence of 68 O-glycosylation sites (in red and numbered).
(TIF)

**S2 Fig. N-glycosylation of rBl-Eng2 is dispensable for Dectin-2 recognition.** SDS-PAGE of nickel purified, full-length rBl-Eng2 (**A**) and ΔC rBl-Eng2 (**B**) treated with PNGase F. The gel was stained with silver nitrate. RNase B was used as control to demonstrate that PNGase F was functional. Red arrows = rBl-Eng2. Black arrows = PNGase F. Green arrows = Rnase B. E = eluate. FL = flow-through. **C)** Purified proteins (E, eluate) were tested in the Dectin-2 reporter assay. The data are representative of three independent experiments.
(TIF)

**S3 Fig. Alignment of full length and truncated recombinant Bl-Eng2 proteins.** The amino acid sequences of the full length, ΔC and ΔN Bl-Eng2 fusion proteins expressed in *Pichia* are shown aligned to illustrate the deleted residues (-). The initiator methione of the native Bl-Eng2 protein has been removed in these constructs. Domains of interest are highlighted in color boxes. Yellow, *Saccharomyces cerevisiae* alpha factor secretory signal peptide sequence; cyan, GH16 domain of Bl-Eng2; red, serine/threonine-rich domain of Bl-Eng2; green, c-Myc epitope tag; black, poly-histidine tag for purification; *, stop codon.
(TIF)

**S4 Fig. Gating strategy for the identification of antigen-specific T cells.** The gating strategy for calnexin- (top), Bl-Eng2- (middle) and NP (bottom) -specific T cells was as follows: FSC and SSC gate was used to eliminate counting beads, singlets were used to eliminate duplets and aggregates; a dump channel composed by CD11c, CD11b, NK1.1 and B220 was used to exclude myeloid cells, NK cells and B lymphocytes; a live stain eliminated dead cells; live cells

were then separated into CD4$^+$ and CD8$^+$ T cells. CD45 mAb was injected intravenously 5 minutes before euthanasia to mark cells in the vasculature.
(TIF)

**S5 Fig. Bl-Eng2 peptide titration for vaccination.** Mice were subcutaneously vaccinated with 10μg ΔN rBl-Eng2 (as adjuvant) emulsified in IFA and with equimolar (0.24 μg) or 10 times more peptide (2.4 μg) relative to the 10 μg full-length rBl-Eng2 (used as control). Two weeks after the vaccine boost, mice were challenged with *B. dermatitidis*. At day 4 post-infection, the frequency and number of total tetramer$^+$ T cells in the lungs (**A**), and IL-17 and IFN-γ producing cells (**B**) were analyzed by flow cytometry. Data shown are from one representative experiment of two performed (n = 5 mice/group). $^*p<0.5$ vs 2.4 μg Bl-Eng2 peptide. **C)** Lung CFU are displayed as the geometric mean with standard deviation. Data are from one representative experiment of two performed (n = 5 mice/group). $^*p<0.05$ vs. 2.4 μg Bl-Eng2 peptide.
(TIF)

**S6 Fig. Deglycosylation of ΔN Bl-Eng2 abolishes its recognition by Dectin-2. A**) SDS-PAGE of ΔN rBl-Eng2 before (red arrow) and after (green arrow) deglycosylation with trifluoro-methanesulfonic acid (TFMS acid). The gel was stained with colloidal Coomassie (blue) and Pro-Q Emerald 300 (gray) to evaluate the molecular weight and the presence of sugar, respectively. **B**) Dectin-2 reporter assay of deglycosylated ΔN rBl-Eng2.
(TIF)

## Acknowledgments

We thank to Dr. Sho Yamasaki (University of Osaka, Japan) who kindly provided the mutant and wild type Dectin-2 reporter cells expressing GFP.

## Author Contributions

**Conceptualization:** Lucas dos Santos Dias, Gregory C. Kujoth, Huafeng Wang, J. Scott Fites, Charles A. Specht, Stuart M. Levitz, Parastoo Azadi, Lingjun Li, Marulasiddappa Suresh, Bruce S. Klein, Marcel Wüthrich.

**Data curation:** Lucas dos Santos Dias, Hannah E. Dobson, Brock Kingstad Bakke, Gregory C. Kujoth, Junfeng Huang, Elaine M. Kohn, Cleison Ledesma Taira, Huafeng Wang, Nitin T. Supekar, Daisy Gates, Christina L. Gomez, Parastoo Azadi, Marcel Wüthrich.

**Formal analysis:** Lucas dos Santos Dias, Hannah E. Dobson, Brock Kingstad Bakke, Gregory C. Kujoth, Junfeng Huang, Elaine M. Kohn, Cleison Ledesma Taira, Huafeng Wang, Nitin T. Supekar, Daisy Gates, Christina L. Gomez, Charles A. Specht, Parastoo Azadi, Lingjun Li.

**Funding acquisition:** Lingjun Li, Marulasiddappa Suresh, Bruce S. Klein, Marcel Wüthrich.

**Investigation:** Lucas dos Santos Dias, Hannah E. Dobson, Brock Kingstad Bakke, Gregory C. Kujoth, Junfeng Huang, Elaine M. Kohn, Cleison Ledesma Taira, Huafeng Wang, Nitin T. Supekar, Parastoo Azadi, Lingjun Li, Marcel Wüthrich.

**Methodology:** Lucas dos Santos Dias, Hannah E. Dobson, Brock Kingstad Bakke, Gregory C. Kujoth, Junfeng Huang, Elaine M. Kohn, Cleison Ledesma Taira, Huafeng Wang, Nitin T. Supekar, Parastoo Azadi, Lingjun Li, Marcel Wüthrich.

**Project administration:** Bruce S. Klein, Marcel Wüthrich.

**Resources:** Lucas dos Santos Dias, J. Scott Fites, Stuart M. Levitz.

**Supervision:** Gregory C. Kujoth, Lingjun Li, Marulasiddappa Suresh, Bruce S. Klein, Marcel Wüthrich.

**Validation:** Gregory C. Kujoth, Junfeng Huang, Elaine M. Kohn, Huafeng Wang, Nitin T. Supekar, Parastoo Azadi, Marulasiddappa Suresh.

**Visualization:** Lucas dos Santos Dias, Hannah E. Dobson, Junfeng Huang, Marcel Wüthrich.

**Writing – original draft:** Lucas dos Santos Dias.

**Writing – review & editing:** Bruce S. Klein, Marcel Wüthrich.

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
