## [Decision Letter · Decision Letter 0]

14 Aug 2020

Dear Marcel:

Thank you very much for submitting your manuscript "Structural basis of Blastomyces Endoglucanase-2 adjuvancy in anti-fungal and -viral immunity" for consideration at PLOS Pathogens. As with all papers reviewed by the journal, your manuscript was reviewed by members of the editorial board and by several independent reviewers. In light of the reviews (below this email), we would like to invite the resubmission of a significantly-revised version that takes into account the reviewers' comments.

The reviewers agreed on the careful nature of the study and quality of the presented data. A novel finding relates to the identification of distinct segments of Blastomyces endoglucanase that contain separable T cell-stimulating and adjuvant properties. However, there were differences in opinion regarding the novelty of the work and its physiologic relevance in vaccine immunity beyond the Blastomyces infection model. I believe that is important for the authors to respond to points raised by reviewer 2, including a virologic readouts in the influenza model. The title should be re-worked to reflect better the findings of the study - at this time, the authors do not have compelling data for an effect on anti-viral immunity outside of stimulating T cell responses.

We cannot make any decision about publication until we have seen the revised manuscript and your response to the reviewers' comments. Your revised manuscript is also likely to be sent to reviewers for further evaluation.

Sincerely,

Tobias M. Hohl

Associate Editor

PLOS Pathogens

Scott Filler

Section Editor

PLOS Pathogens

Kasturi Haldar

Editor-in-Chief

PLOS Pathogens

orcid.org/0000-0001-5065-158X

Michael Malim

Editor-in-Chief

PLOS Pathogens

orcid.org/0000-0002-7699-2064

The reviewers agreed on the careful nature of the study and quality of the presented data. A novel finding relates to the identification of distinct segment of Blastomyces endoglucanase that contain separable T cell-stimulating and adjuvant properties. However, there were differences in opinion regarding the novelty of the proposed work and its physiologic relevance beyond the Blastomyces infection model. I believe that is important for the authors to respond to points raised by reviewer 2, including a virologic readouts in the influenza model. The title should be re-worked to reflect better the findings of the study - at this time, the authors do not have compelling data for an effect on anti-viral immunity outside of stimulating T cell responses.

Reviewer's Responses to Questions

**Part I - Summary**

Reviewer #1: In this study by Dos Santos Dias et al., the authors determine the molecular basis of the adjuvant effect of a Dectin-2 agonist that they previously identified in Blastomyces. The report lines up with a continuous series of elegant studies from the Wüthrich/Klein group on T cell immunity against the dimorphic fungi, including the identification of antigens and PAMPs.

While they have previously identified endoglucanase 2 to trigger Dectin-2 signalling, and later to comprise a T cell epitope, they now characterize the Dectin-2 agonist activity in detail and localize it to O-linked mannans in the C-terminus of the glycoprotein. They further show that the C-terminal part of the endoglucanase is sufficient to trigger T cell expansion, Th1 and Th17 differentiation and antifungal effector functions in the infected lung in a Dectin-2 and glycosylation-dependent manner. Finally, they show that the endoglucanase-2 C-terminus also mediates adjuvant activity in combination with a viral antigen and promotes immunity against IAV. Although adjuvant effect of DeltaN-rBl-Eng2 on the antiviral response is limited and only adds a small increment over the effect by Adjuplex, the data show that the newly characterized fungal adjuvant can boosts responses against agents other than fungi, thereby highlighting the potential of the adjuvant for vaccine development. The study is carefully designed and well performed. I only have a few minor comments.

Reviewer #2: In this article by Dos Santos Dias, et al the authors build upon previous studies in their lab that demonstrated an important role for endonuclease-2 from Blastomyces (Bl-Eng2) as adjuvant promoting antifungal immunity via engagement of host Dectin-2. In follow up work they furthered identified an immunodominant T cell epitope within the Bl-Eng2 gene and that T cells specific against this epitope could confer vaccine-induced protection upon challenge with full length Bl-Eng2. Thus, the contributions of full length Bl-Eng2 to the activation of antifungal immunity are due to potentially distinct functions as both antigen and adjuvant. In the current study the investigators perform a series of detailed analyses that lead them to identify the specific segments of Bl-Eng2 that act as adjuvant. They show that the adjuvant segment is distinct from the T cell epitope region. They then follow up on their previous work and further show that a truncated version of Bl-Eng2 (devoid of T cell epitope), retains potent adjuvant activity in a Dectin-2 dependent manner. They further show that the adjuvant activity of truncated Bl-Eng2 requires glycosylation, specifically O-mannosylation in order to trigger Dectin-2-dependent signaling and downstream enhancement of cytokine production and T cell activation and differentiation of antifungal T cells. The authors go on to further test whether the adjuvant activity of Bl-Eng2 can function beyond boosting antifungal responses. A mixture of Eng2 with influenza antigen failed to activated virus-specific T cells. In contrast, a complex of truncated Bl-Eng2 together with influenza nucleoprotein and adjuplex promoted the enhanced activation and differentiation of polyfunctional flu-specific T cells. Overall the studies are carefully performed and well controlled.

Reviewer #3: This diligently conducted study investigated the adjuvant properties of the Blastomyces endoglucanase-2 (Bl Eng-2) protein, a dectin-2 ligand. By using recombinant proteins or chemical deglycosylation strategies, the authors convincingly demonstrate that the adjuvant properties lie within the c-terminus glycosylation structure, rather than the protein backbone, and through binding to dectin-2. More specifically, o-linked glycans and not N-linked glycans, appear to be responsible for this adjuvant properties. These adjuvant properties induce specific CD4+ and CD8+ T cell immune response against fungal (Blastomyces) or viral (influenza virus) pulmonary challenge in mice. This novel adjuvant adds to a new generation of adjuvants that are critically needed for developing enhanced vaccines, especially those requiring a skewed immune response towards a specific Th1 and Th17 immune polarization. The study is also meticulously conducted with proper controls ad convincing data. I have minor comments below.

**Part II – Major Issues: Key Experiments Required for Acceptance**

Reviewer #1: none

Reviewer #2: 1) The addition of Bl-Eng2 to adjuplex and NP appears to promote enhanced flu-specific T cell responses in terms of numbers for antigen-specific T cells detected in the lung and cytokine production by these cells. Have the authors tested whether this response results in superior protection against challenge with Influenza in terms of reductions in viral titers and survival from infection?

2) Conceptually, one would predict that Bl-Eng2 would be an excellent adjuvant to boost immunity against various clinically relevant fungi but not necessarily other types of pathogens. Influenza infections are certainly of significant clinical importance but focusing the analysis of Bl-Eng2 as adjuvant in this model seems like a random choice and lost opportunity. Their own data shows that Bl-Eng2 basically had no adjuvant activity for the activation of antiviral T cells and NP+Bl-Eng2 failed to activate any T cell responses above background of NP alone (A finding contrary to the stated title suggesting that Eng-2 is an adjuvant for antiviral immunity). Furthermore, without the protection data asked in point 1 we do not know if the statistically significant enhancements in T cell response induced by addition Bl-Eng2 to Adjuplex are biologically significant to antiviral defense. Perhaps Bl-Eng2 is a potent but restricted antifungal adjuvant. Have the authors tested whether truncated Bl-Eng2 can function as an adjuvant to boost superior immunity against fungal pathogens other than Blastomyces?

3) A lot of the exciting data on Eng2 as novel adjuvant via activation of Dectin-2 have been previously reported in excellent publications by this team of investigators including previous PLoS Pathogens and Mucosal Immunology papers. The novel data presented in the current paper appears to be a narrowing down of adjuvant activity of the Eng2 gene to the C-terminus and need for specific glycosylation signals. Perhaps I have missed other important, new findings in this paper? A further identification of the extent of truncated Eng2 adjuvant activity against a diversity of fungi might help broaden the impact of the current study.

Reviewer #3: None needed

**Part III – Minor Issues: Editorial and Data Presentation Modifications**

Reviewer #1: Minor comments:

1. References 23 and 27 are the same.

2. In the introduction, where the authors talk about vaccine-induced T cell immunity against fungi, they only refer to their own work on dimorphic fungi (which is undoubtedly of great relevance), but they should also include work from other fungi, e.g. C. albicans Als3.

3. Figure 1A shows >2 bands for the E. coli-expressed rBl-Eng2, without the authors commenting on it.

4. Figure 6: In some figure panels, certain symbols look like they could be outlayers (e.g. Fig 6B, NP366 and NP311-specific events in the NP + DeltaN-rBl-Eng2 + Adjuplex-treated group (Parenchyma); Fig. 6DNP45 47 52-specific IFNg production in the NP + DeltaN-rBl-Eng2 + Adjuplex-treated group). Did they authors perform an outlayer analysis? In case, outlayers are found, they should be excluded and statistical significances re-calculated.

5. Supplementary Figure 5: Authors show the statistical significance (resp. the non-significance) for the difference between the last to groups. What about the difference between FL rBl-Eng2 versus the combination of the BL-Eng2 peptide + DletaN-rBL-Eng2 (which is at the centre of what this experiment addresses)?

6. Supplementary Figure 6B: the panel lacks a y-axis label.

Reviewer #2: (No Response)

Reviewer #3: 1) The authors show that transgenic mice expressing CD4+ T cells with TCR specific for protective epitope/shared antigen for Blastomyces dermatitidis, Histoplasma capsulatum, Coccidioides posadasii, Paracoccidioides lutzii, and Paracoccidioides brasiliensis are protected via a Th1/Th17 immune response in the lungs. Do they expect similar outcome with vaccines targeting other fungal infections outside the above-mentioned pathogens (e.g. Cryptococcus or Candida) and known to require similar mechanism of protection?

2) In figure 1, Can the authors comment on the inconsistency in the size of the rBl-Eng2 expressed in P. pastoris in A (between 75-100 kDa) and C (between 100-150 kDa). Also, the E. coli-expressed Bl-Eng2 is not pure. Is it possible the impurities are blocking the activation of the reporter cells?

3) On page 16 (in their attempt to investigate if Bl-Eng2 stimulated immunity via C-terminus O-mannans or due to N-terminus CD4+ T cell antigen) and to make it clear, it is best to add a sentence to explain that the adoptively transferred, congenic (CD90.1+) TCR transgenic mice has been vaccinated with Bl-Eng2 combined with each of the tested antigen. Perhaps the sentence can be rephrased as “First, we formulated the immunodominant T cell epitope of calnexin (35) with ∆N rBl-Eng2, vaccinated and then measured T cell expansion and differentiation of adoptively transferred, congenic (CD90.1+) TCR transgenic 1807 cells (SFig. 4).

4) Similarly, to enhance clarity and at the bottom of page 16, it would be better to mention reduction in B. dermatitidis lung CFU.

5) In Fig 4, marking the flow cytometry data panels in C (i.e. lungs, vasculature, and IL 17 and IFN-� cells (in lungs??)) would also help.

6) If Bl-Eng2 and/or the N-terminus truncated protein are proposed as adjuvants, why is it being coupled with IFA? Is this for a depot effect? Have the authors tried Bl-Eng2 without IFA?

7) Is there an explanation why a higher dose of the Bl-Eng2 peptide is less immunogenic and protective against B. dermatitidis?

8) How long is the vaccine effect expected to last and is a boost dose required or a single dose would result in a protective immune response?

9) The data for vaccinating Dectin2-/- mice with Bl-Eng 2 peptide and Degly ∆N rBl-Eng2 is missing from Fig 5. I understand that the expected data would be negative, but would be reassuring to have this extra control for these studies.

PLOS authors have the option to publish the peer review history of their article (what does this mean?). If published, this will include your full peer review and any attached files.

Reviewer #1: No

Reviewer #2: No

Reviewer #3: **Yes: **Ashraf Ibrahim
---

## [Editor Report · Decision Letter 1]

19 Jan 2021

Dear Marcel,

We are pleased to inform you that your manuscript 'Structural basis of Blastomyces Endoglucanase-2 adjuvancy in anti-fungal and -viral immunity' has been provisionally accepted for publication in PLOS Pathogens.

Best regards,

Tobias M. Hohl

Associate Editor

PLOS Pathogens

Scott Filler

Section Editor

PLOS Pathogens

Kasturi Haldar

Editor-in-Chief

PLOS Pathogens

orcid.org/0000-0001-5065-158X

Michael Malim

Editor-in-Chief

PLOS Pathogens

orcid.org/0000-0002-7699-2064

The authors have responded to the reviewers' comments in a comprehensive manner and have added new data that strengthens the novelty and significance of their findings.
---

## [Editor Report · Acceptance letter]

24 Feb 2021

Dear Dr. Wuthrich,

We are delighted to inform you that your manuscript, "Structural basis of Blastomyces Endoglucanase-2 adjuvancy in anti-fungal and -viral immunity," has been formally accepted for publication in PLOS Pathogens.

Best regards,

Kasturi Haldar

Editor-in-Chief

PLOS Pathogens

orcid.org/0000-0001-5065-158X

Michael Malim

Editor-in-Chief

PLOS Pathogens

orcid.org/0000-0002-7699-2064